# Auditory Processing and Speech Sound Disorders: Behavioral and Electrophysiological Findings

**DOI:** 10.3390/audiolres15050119

**Published:** 2025-09-19

**Authors:** Konstantinos Drosos, Paris Vogazianos, Dionysios Tafiadis, Louiza Voniati, Alexandra Papanicolaou, Klea Panayidou, Chryssoula Thodi

**Affiliations:** 1School of Sciences, Speech and Language Therapy, European University Cyprus, 2404 Nicosia, Cyprus; l.voniati@euc.ac.cy (L.V.); c.thodi@euc.ac.cy (C.T.); 2School of Humanities, Social and Education Sciences, Department of Social and Behavioral Sciences, European University Cyprus, 2404 Nicosia, Cyprus; p.vogazianos@euc.ac.cy; 3Department of Speech and Language Therapy, University of Ioannina, GR-45500 Ioannina, Greece; tafiadis@uoi.gr; 4Department of Hearing and Speech Sciences, University of Maryland College Park, College Park, MD 20740, USA

**Keywords:** auditory processing, speech sound disorders, phonological assessment, electrophysiological observation, ABR, correlations

## Abstract

**Background:** Children diagnosed with Speech Sound Disorders (SSDs) encounter difficulties in speech perception, especially when listening in the presence of background noise. Recommended protocols for auditory processing evaluation include behavioral linguistic and speech processing tests, as well as objective electrophysiological measures. The present study compared the auditory processing profiles of children with SSD and typically developing (TD) children using a battery of behavioral language and auditory tests combined with auditory evoked responses. **Methods:** Forty (40) parents of 7–10 years old Greek Cypriot children completed parent questionnaires related to their children’s listening; their children completed an assessment comprising language, phonology, auditory processing, and auditory evoked responses. The experimental group included 24 children with a history of SSDs; the control group consisted of 16 TD children. **Results:** Three factors significantly differentiated SSD from TD children: Factor 1 (auditory processing screening), Factor 5 (phonological awareness), and Factor 13 (Auditory Brainstem Response—ABR wave V latency). Among these, Factor 1 consistently predicted SSD classification both independently and in combined models, indicating strong ecological and diagnostic relevance. This predictive power suggests real-world listening behaviors are central to SSD differentiation. The significant correlation between Factor 5 and Factor 13 may suggest an interaction between auditory processing at the brainstem level and higher-order phonological manipulation. **Conclusions:** This research underscores the diagnostic significance of integrating behavioral and physiological metrics through dimensional and predictive methodologies. Factor 1, which focuses on authentic listening environments, was identified as the strongest predictor. These results advocate for the inclusion of ecologically valid listening items in the screening for APD. Poor discrimination of speech in noise imposes discrepancies between incoming auditory information and retained phonological representations, which disrupts the implicit processing mechanisms that align auditory input with phonological representations stored in memory. Speech and language pathologists can incorporate pertinent auditory processing assessment findings to identify potential language-processing challenges and formulate more effective therapeutic intervention strategies.

## 1. Introduction

Auditory processing encompasses the ability and efficiency of the central nervous system to interpret auditory information, enabling individuals to identify sound sources, recognize and comprehend auditory stimuli, discriminate between sounds, and concentrate on receiving information in both quiet and noisy environments [1,2]. As the auditory nervous system develops, we observe the refinement of complex auditory functions like sound localization, auditory discrimination, temporal auditory processing, dichotic listening, sound pattern recognition, and the ability to interpret distorted speech or speech in background noise [3,4,5]. Language and phonological skill development depend heavily on the integrity of auditory perception and processing [6,7,8]. Auditory Processing Disorders (APDs) denote the brain’s inability to effectively process auditory signals, even when accurate information is received, and can lead to difficulties in understanding and remembering auditory stimuli, particularly verbal language [2,9]. In pediatric populations, the prevalence of APD is estimated to be between 0.2% and 5% [9,10,11], with a diagnosis ratio of 2:1 for males over females.

Imhof [12] noted that 60% of the instructional time allocated to primary school children is spent on listening activities. Consequently, APD may adversely affect academic performance [13,14,15,16]. The relationship between auditory processing and language development has been investigated extensively in the literature [9,17,18,19]. A key element of auditory processing, namely auditory discrimination, fosters the ability to distinguish between phonemes. Auditory discrimination is essential for accurate understanding, phoneme acquisition, and effective verbal communication [16]. Children diagnosed with APD frequently encounter challenges with auditory figure-ground, auditory closure, binaural processing, temporal processing, and auditory memory [20,21,22]. Conversely, children exhibiting atypical speech patterns demonstrate deficits in temporal resolution, sound localization, sequential sound memory, speech discrimination in noise, and auditory closure [2]. APD may impede language acquisition, and phonological skill development, and may result in Speech Sound Disorder (SSD) and difficulties with reading and writing [13,23,24,25,26,27]. Research has consistently shown a high rate of comorbidity and a significant link between APD and SSD [21,27].

Children diagnosed with APD exhibited performance levels significantly lower than Typically Developing (TD) children: they displayed syllable and phoneme identification and discrimination task deficits in both challenging (noisy) and nonchallenging (quiet) conditions in [28,29,30]. This difficulty implied a compromised neural mechanism extending from the subcortical to the cortical level [31]. Disruptions or deficiencies in cortical auditory processing may greatly impede the integration, comprehension, and interpretation of sound stimuli [31,32]. Specifically, children with SSD face difficulties with auditory feedback, which is essential for perceiving sounds [33]. APD can destabilize phonemic representations in the brain and hinder speech perception, making the acquisition of phonology and language more challenging [34,35,36,37]. High comorbidity rates between auditory processing and language, and SSD have been documented in the literature [14,27,36,38]. Children with APD often exhibit more severe phonological disorders [36,38,39]; they tend to take longer to respond during tasks that require auditory analysis and synthesis, and they face difficulties with temporal ordering and memory retention [40]. Children with phonological disorders exhibit poorer auditory perceptual accuracy in lexical and phonemic judgments compared to children who develop typically [14,41,42].

The assessment of APD in children is complicated by the possible overlap of auditory processing, language, and cognitive deficits, as there is no standardized test battery currently available to identify the underlying causes of their difficulties [27,43,44]. The recommended assessment protocol includes initially an audiological evaluation featuring a pure tone audiogram, recognized as the gold standard for hearing assessment [13,45] along with tympanometry and word discrimination tasks in both quiet and noise, followed by temporal frequency and duration pattern recognition, and dichotic listening. A comprehensive language evaluation including phonology, memory, and attention tasks is part of the protocol, to highlight areas for intervention [44,46,47,48]. The auditory processing assessment protocol is completed with electrophysiology [49].

Electrophysiological assessment targets the neurobiological processes underlying speech perception in noise and the operational dynamics within the auditory system [50,51]. The click-evoked Auditory Brainstem Response (ABR) identifies fundamental neural irregularities and is instrumental in examining the integrity of auditory pathways [49]. The ABR provides an objective measure in the assessment of patients unable to respond consistently during behavioral audiological tests [50]. Objective measures mitigate the influence of attention, motivation, mood, responsiveness, cognition, and other variables during auditory evaluations [52]. Evidence of APD has been documented with electrophysiology indices: children diagnosed with APD had significantly longer latencies in speech-evoked ABRs compared to those of TD children [49]. Click-evoked ABR in children with or at risk for APD showed prolonged latencies at high presentation rates and reductions in wave I, III and V amplitudes, compared to control groups [53,54]. Cortical auditory evoked potentials (CAEPs) are acknowledged as neurophysiological markers of speech processing capabilities [55]. From infancy to adolescence the N1-P2 complex evolves to become increasingly distinct [56] and there is a reduction in the latency of P1 and a significant decrease in the amplitudes of waves P1 and N2. Waves P1 and N2 have been linked to discrete dimensions of auditory processing, as P1 is associated with auditory features such as frequency, intensity, and timing, and N2 is related to the transformation of this information into sensory representations [55,56]. Late Latency response amplitudes in children diagnosed with APD were significantly different from those of age-matched control subjects, whereas there was no notable difference in latencies [57,58].

The purpose of our study was to investigate the auditory processing profiles of children with SSDs and TD children, using a battery of behavioral language and auditory tests combined with electrophysiology measures. The comprehensive assessment battery aimed to investigate the sensitivity, validity, and correlations between assessment indices and to identify areas for AP intervention.

## 2. Materials and Methods

### 2.1. Sample

The sample size in our study was calculated utilizing the EPI INFO 7 software developed by the WHO for the field of psychiatry, with a confidence interval of 95% and an expected SSD prevalence of 1.7% among the school-aged population as indicated in the Cyprus Ministry of Education, Youth and Sports in the yearly report of 2016., and an accuracy of 3%. The sample size calculation yielded a minimum of 14 children per group.

Forty Greek Cypriot children between seven and ten years old participated in this study. Sixteen children who exhibited typical psychosocial and language development, of average age 8.40 years (7.85–8.70 years), formed the typically developing (TD) group. Twenty-four children with a history of diagnosis and intervention for Speech Sound Disorders (SSD), of average age 8.1 years (7.90–8.40 years) formed the experimental group [59]. All children completed the audiometry, behavioral language and auditory processing tests. Electrophysiology testing was completed by fourteen TD children of average age 8.40 years (7.85–8.70 years), and fourteen children with a history of SSD diagnosis and average age of 8.1 years (7.90–8.40 years). All children were monolingual Greek Cypriot dialect speakers, had normal pure-tone thresholds, no history of middle ear disease, and no sensory, cognitive, psychomotor or learning disorders.

Implementation of the test battery took place over three distinct sessions. During the initial meeting, evaluations of hearing, language, and phonology were conducted, while parents completed two screening questionnaires related to auditory processing, CHAPS and APDQ [59]. Subsequently, an audiologist administering tests designed to assess auditory processing abilities following the ASHA [46] and AAA [60] guidelines as well as literature recommendations for a complete auditory processing evaluation [13,23,32,44,45,60,61]. The third session consisted of the electrophysiological assessment [49,62,63]. Findings from the first two parts of the assessment protocol were reported by Drosos et al. [59] for 24 participants in the group of children with SSD and 16 participants in the TD group. We present here findings for 14 age-matched children in each group: the participant numbers were defined by a lower completion rate for the electrophysiological assessment.

### 2.2. Participant Recruitment and Language and Auditory Processing Assessment

Approval for the recruitment strategy and consent form was obtained from the Cyprus National Bioethics Committee (EEBK/EP/2022/37). All parents of children undergoing therapy for Speech Sound Disorders (SSDs) at the Speech, Language, and Hearing Clinic of the European University Cyprus were invited to participate in the study. Parents who agreed to their children’s participation completed and signed the consent form. Additionally, they provided information about their children’s health histories. Children with typical development (TD) were recruited from public schools; their inclusion was approved by the Pedagogical Institute of the Ministry of Education, Sport, and Youth of the Republic of Cyprus (KEEA-148053). The behavioral test protocol was also presented in [59]. For the sake of completeness, we present the protocol here. A pure-tone audiogram was administered by an audiologist at the Speech, Language, and Hearing Clinic of the European University Cyprus [64]. Consequently, the children completed the auditory processing evaluation comprising the following tests: Gap Detection [65], Gaps-in-Noise [66], Dichotic Hearing (words) [67], the Greek Speech in Babble [68], Duration and Frequency Pattern Sequence [68], and Forward/Backward digit span [69]. Gap Detection and Gaps in Noise incorporate non- linguistic stimuli, following the implementation protocol established by Musiek [68]. The Dichotic Listening and Speech in Babble test administered has been validated for the Greek language [23,67]. Gap detection entails detecting a short silence or pause between two sounds within a continuous auditory stream [68]. Gaps in Noise refers to the ability to detect a brief silent interval or gap within a background of continuous noise. Gap detection in noise is often used in auditory processing research to evaluate how well individuals can perceive temporal cues when they are masked by ongoing, potentially distracting background sounds [68]. Dichotic listening (words) involves the simultaneous presentation of different words to each ear. The goal is to assess an individual’s ability to process and integrate sounds coming from both ears, as well as to evaluate their auditory attention and lateralization abilities. Dichotic tasks assess how well each hemisphere of the brain processes auditory information and show the hemispheric specialization and cross-communication [23,68].

The behavioral test protocol also included a series of speech and language evaluations, namely the Action Picture Test (APT) [70] and the full version of the Metaphone test (assessment of Phonological Development and Reading Readiness in Phonological Awareness) [71]. The APT evaluates the content and grammatical structures present in children’s speech, providing insights into their morphological development in grammar and syntax. This assessment considers the grammatical forms and vocabulary used for effective communication. The Metaphone test evaluates linguistic skills related to rhyme, syllable, and phoneme awareness. Additionally, the Raven Colored Progressive Matrices Assessment (Raven) [72] was included in the evaluation protocol to measure cognitive performance. The Raven is a non-verbal tool that evaluates abstract reasoning, cognitive function, spatial reasoning, analogical reasoning, and problem-solving capabilities.

### 2.3. Electrophysiological Assessment

The Vivosonic Integrity v500 two-channel evoked potential system was employed to capture the auditory evoked potentials [73]. Before electrode placement, the skin was treated with NuPrep, an abrasive gel designed to reduce electrode impedance to less than 5 kΩ. Subsequently, electrodes were positioned in a vertical electrode configuration (from the ipsilateral and contralateral mastoids to high forehead Fz) with the ground electrode placed in the lower forehead. The evoked responses were amplified and filtered to capture the early, middle, and late auditory responses, according to published guidelines. Recording parameters for the electrophysiological assessment as shown in Table 1.

Developmental reports on MLR and LLR indicate that auditory cortex maturation manifests as modified hemispheric asymmetry. The two hemispheres may support distinct auditory processing specializations: these asymmetries were assessed with ipsilateral and contralateral AEP recordings. Ipsilateral AEP reflects electrical activity in response to auditory stimuli delivered to the same ear; latencies and amplitudes were calculated for both the ipsilateral and contralateral recordings of ABR, MLR and LLR. This methodology has been applied to evaluate the overall functionality of the auditory system, with a particular focus on the integrity of the neural pathways involved [22,29].

ABR absolute and inter-wave latencies (ms) and amplitudes (uV) were determined for ipsilateral and contralateral recordings. ABR analysis included identification of waves I (only for ipsilateral recordings), III, and V, with latencies, interpeak interval (I–V) for the ipsilateral responses, and amplitudes. Waves Pa and Na of MLR were noted. For LLR, we identified waves P1, N1, P2, and the interweaves of P1-N1 and N1-P2.

### 2.4. Statistical Analysis

Dimension reduction and exploratory factor analysis were employed to extract meaningful dimensions in our data. Fifteen dimensions were extracted with all determinants being greater than zero, the Kaiser-Meyer-Olkin Measure of Sampling Adequacy (KMO) values ranging between 0.500 and 0.756, the Bartlett test of sphericity for all factors extracted being significant at *p* < 0.001, the cumulative variances ranging between 40% and 94% and the weights of items in the factors ranging between 0.604 and 0.970. An overall value was calculated for the fifteen dimensions using the factor loadings of the items, and the total scores were tested for normality using the Shapiro–Wilk test since *n* < 50, with normality rejected where results were *p* < 0.05. Differences in performance between TD and APD groups in scales for which the Shapiro–Wilk test did not reject normality were assessed with t tests for independent samples, whereas differences in performance in scales for which the Shapiro–Wilk test rejected normality were assessed with the Mann–Whitney U test for independent samples.

Binary logistic regression was employed to assess the effect of the factors, as it does not assume normality of variables. Binary logistic regression assumes that: the dependent variable is dichotomous; observations are independent from each other; there is no perfect multicollinearity between independent variables; and that there is a linear relationship between any numerical independent variables and the logit transformation of the dependent variable [74]. These assumptions were tested using the Box-Tidwell transformation [75]. In models where independent variables were not statistically significant, parameters from simpler models were used to calculate the minimum sample size that would be needed to achieve significance. Acceptable significance levels were *p* < 0.05, two-tailed. The statistical analyses were performed using SPSS version 29 [76].

## 3. Results

### 3.1. Summary of Demographic and Behavrioral AP Assessment Findings

Participants demographics and language performance for the entire sample have been presented in Drosos et al. 2024b [59]. This report includes an assessment of correlations between the behavioral and physiological findings. In the Appendix A, the latencies of ABR, MLR, and LLR are provided, along with the corresponding waveforms for both groups. Appendix B contains detailed data on the performance of participants from the two groups in electrophysiological assessments.

Significant differences between the two groups in language and AP tests were reported in Drosos et al. 2024b [59]. Children with SSD had significantly reduced performance in:-The Speech in Babble test [67] at S/N form −1 and 3 (with the −1 condition being when the noise level exceeds that of speech by 5 dB).-Temporal processing, frequency perception, and dichotic listening tests from the auditory processing assessment battery.-Grammar, specifically the use of prepositional phrases from the language battery.-Immediate word repetition and initial phoneme deletion from the phonology battery.

### 3.2. Group Comparisons

To comprehensively analyze questionnaire, behavioral, and electrophysiological data, we implemented dimension reduction through exploratory factor analysis. The factors extracted through factor analysis were tested for significant differences between the experimental group and the control group: three factors were identified. Factor 1—CHAPS included performance on CHAPS (CHAPS NOISE + CHAPS QUIET). Factor 5—PHONOLOGY reflected overall performance in phonology (Rhyme Identification, First Syllable Localization, First Phoneme Identification, Word Finding, Phoneme, Word Repetition). Factor 13—ABR-wV consisted of ABR ipsilateral wave V latencies in the Right and Left Ears.

The Shapiro–Wilk test for normality indicated that normality was not rejected for Factor 1 (W = 0.962, *p* = 0.392) as well as for Factor 13 (W = 0.969, *p* = 0.556). Conversely, the test did reject normality for Factor 5 (W = 0.906, *p* = 0.016). Factor 1 exhibited a KMO of 0.657, with Bartlett’s test yielding *p* < 0.001, explaining a total variance of 40% and factor loadings exceeding 0.630. Factor 5 demonstrated a KMO of 0.756, with Bartlett’s test also yielding *p* < 0.001, accounting for 59% of the total variance and factor loadings greater than 0.73. Lastly, Factor 13 had a KMO of 0.500, with Bartlett’s test indicating *p* < 0.001, explaining 78% of the total variance and factor loadings above 0.84. Table 2 shows the factors’ internal consistency coefficient, Cronbach’s α. Given the sample size and the number of items employed, Factor 5 and Factor 13 had excellent internal consistency, whereas Factor 1 had fair internal consistency [77].

Statistically significant differences t(26) = −2.48, *p* = 0.010 were identified in Factor 1 between TD (M = 0.3714, SD = 0.3338) and APD (M = 0.6357, SD = 0.2196) which, due to Levene’s test of equality of variances not rejecting homogeneity (*p* = 0.083), was calculated assuming equal variances. Statistically significant differences t(26) = 2.32, *p* = 0.014 were also identified in Factor 13 between TD (M = 0.4232, SD = 0.0973) and SSD (M = 0.3336, SD = 0.1067) which, due to Levene’s test of equality of variances not rejecting homogeneity (*p* = 0.781), was calculated assuming equal variances. Finally, statistically significant differences U = 53, *p* = 0.039 were also identified in Factor 5 between TD (Median = 22.30, IQR = 3) and APD (Median = 20.40, IQR = 1.40).

#### Binary Logistic Regression and Correlations

Binary logistic regression indicates that Factor 1 is a significant predictor of group differentiation, with [χ^2^ = 5.754, df = 1 and *p* = 0.016]. Factor 1 had a pseudo R^2^ = 0.186 whilst it was significant [Wald = 4.539, *p* = 0.033]. The odds ratio (OR) was 29.276 (95%CI 1.310–654.094) and the model correctly predicted 71.4% of the cases. Binary logistic regression indicates that Factor 5 is also a significant predictor of group differentiation, with [χ^2^ = 5.111, df = 1 and *p* = 0.024]. Factor 5 had a pseudo R^2^ = 0.167, and it was significant [Wald = 4.129, *p* = 0.042]. The odds ratio (OR) was 0.577 (95%CI 0.339–0.981) and the model correctly predicted 67.9% of the cases. Finally, binary logistic regression indicates that Factor 13 is a significant predictor of group differentiation with [χ^2^ = 5.209, df = 1 and *p* = 0.022]. Factor 13 had a pseudo R^2^ = 0.170 and was significant [Wald = 4.094, *p* = 0.043]. The odds ratio (OR) was 1.0 × 10^−4^ (95%CI 2.0 × 10^−8^–0.753) and the model correctly predicted 60.7% of cases. A summary of these findings is shown in Table 3.

Correlation between Factor 1—CHAPS, Factor 5—PHONOLOGY and Factor 13—ABR wV using Spearman’s and Pearson’s correlation coefficients indicated that Factor 1—CHAPS and Factor 13—ABR wV were not correlated rp = −0.141, *p* = 0.475, Factor 1—CHAPS and Factor 5—PHONOLOGY were not correlated rs = −0.251, *p* = 0.197 whereas Factor 5—PHONOLOGY and Factor 13- ABR wV were correlated rs = 0.385, *p* = 0.043. As Factor 5—PHONOLOGY and Factor 13—ABR wV were significantly correlated, they were not included together in a model. Table 4 shows a summary of the correlation analysis, and Table 4 shows the binary logistic regression for the 3 factors.

When Factor 1—CHAPS and Factor 5—PHONOLOGY were included together in a binary logistic regression model, the model was significant [χ^2^ = 10.200, df = 2, and *p* = 0.006] with a pseudo R^2^ = 0.305. Neither factor was individually significant: Factor 1—CHAPS [Wald = 3.839, *p* = 0.050; Factor 5—PHONOLOGY [Wald = 3.658, *p* = 0.056]. Power analysis using G*Power,3.1.9.7 versionassuming a single tail, a null hypothesis of Pr(Y = 1|X = 1) H0 = 0.57, an alternative hypothesis Pr(Y = 1|X = 1) H1 = 0.79, a correlation between the independent variables of 0.25, a statistical significance of 0.05 and a statistical power of 0.8 indicated that the parameters of the model were significant with a minimum sample size of 38.

When Factor 1—CHAPS and Factor 13—ABR wV were included together in a binary logistic regression model, the model was significant χ^2^ = 10.762, df = 2, and *p* = 0.005 with a pseudo R^2^ = 0.319. Factor 1—CHAPS was statistically significant [Wald = 4.195, *p* = 0.041], whereas Factor 5—PHONOLOGY was not statistically significant [Wald = 3.720, *p* = 0.054]. The odds ratio (OR) for Factor 1—CHAPS was 46.503 (95%CI 1.180—1832.899) and although Factor 13—ABR wV was not significant, the model correctly predicted 75.0% of the cases. Power analysis using G*Power, and assuming a single tail, a null hypothesis of Pr(Y = 1|X = 1) H0 = 0.57, an alternative hypothesis Pr(Y = 1|X = 1) H1 = 0.79, a correlation between the independent variables of 0.14, a statistical significance of 0.05 and a statistical power of 0.8 indicated that the parameters of the model were significant with a minimum sample size of 36.

### 3.3. Receiver Operating Characteristics (ROC) Analysis for Auditory Processing Tests

To evaluate the performance of our binary classification model and illustrate the trade-off between the true positive rate (sensitivity) and the false positive rate at various decision thresholds, the ROC curve plots were constructed for significant factors. The decision thresholds and the area under the curve (AUC) provide a single metric summarizing overall model performance. A summary of the Receiver Operating Curve analysis for our significant factors is presented in Table 5.

Figure 1, Figure 2 and Figure 3 illustrate the ROC analysis for each significant factor; Figure 4 shows the ROC analysis for the Factor 1 and Factor 13 together.

## 4. Discussion

This study examined the auditory processing characteristics of Greek Cypriot children aged 7–10 years with a history of SSD and compared findings to age-matched typically developing controls. We identified differences in auditory processing indices between the two groups and investigated correlations between language and auditory processing with behavioral and electrophysiological measures. In this discussion section, we compare our auditory processing findings in children with a history of SSD to the literature referring to children diagnosed with/or suspected of APD. Thus, we recommend a cautionary approach when relating our findings to children with APD diagnosis.

Recent systematic reviews suggested that AP difficulties may be a significant characteristic in children with SSDs, as evidenced by frequent reports of lower scores on auditory processing tasks among children with SSDs [19,27]. Children with phonological disorders scored at or below the 10th percentile in the Speech and Auditory Processing (SAP) subscale of the Evaluation of Children’s Listening and Processing Skills (ECLiPS) [16]. Furthermore, children with phonological disorders had an odds ratio of 2.8 for experiencing challenges in auditory processing skills and listening difficulties compared to TD children [16,78]. This study revealed significant differences between the group of children with a history of SSD and the group of TD children. Statistical analyses identified three primary factors (Factor 1: CHAPS, Factor 5: PHONOLOGY, Factor 13: ABR Wv) as distinguishing characteristics between the two groups. These factors demonstrated strong predictive capability in binary logistic regression models. The low average score of the experimental group on Factor 5—PHONOLOGY indicates weaknesses in fundamental phonological skills, thereby supporting the theory of phonological deficiency as central to the language difficulties observed in children with SSD [79]. The differences noted in Factor 1—CHAPS further reinforce the notion that children with APD exhibit diminished functional auditory perception, particularly in noisy environments, as highlighted by Keith et al. [62] and Dawes & Bishop [80]. When these auditory processing difficulties are not properly diagnosed, children with SSDs may have a less favorable prognosis in therapy. Our research found auditory processing indices that may lead to more personalized and accurate diagnosis of auditory function in children with SSDs and thus inform effective interventions.

### 4.1. Receiver Operative Characteristics (ROC) for Auditory Behavioral Scores

The literature on ROC curve presentation highlights the need for specialized cut-off scores for TD children, diagnosed populations with APD, and populations with comorbid APD, to replace the assumption estimating that a performance below 2 standard deviations lower than the mean signifies a disordered score [81]. Sanchez and Lam [82] reported cut-off scores on ROC curves for the Competing Sentences and Speech in Babble tests, for children with Attention Deficit/Hyperactivity Disorder (ADHD). The ROC analyses of this study indicated cut-off values and ROC curves for a series of auditory processing indices, including CHAPS, Phonology, ABR wave I–V interwave latency, and a combination of CHAPS and I–V latency, indicating the presence of auditory processing difficulties among children with SSD. There are differences between our study and Sanchez and Lam [82], in terms of sample and statistical methodology, as well as cut-off level calculation methods. Nevertheless, it is evident that in both studies, auditory processing indices are effective predictors for difficulties experienced among children with SSD, even after completion of speech-language treatment.

### 4.2. Electrophysiological Assessment and Auditory Processing Correlations

This is the first study to assess ABR, MLR, and LLR measures and linguistics concurrently with auditory processing tests in children with SSD. The cautionary approach when relating findings in children with a history of SSD to those of children suspected of or diagnosed with APD is strongly indicated when interpreting electrophysiological findings in the SSD group.

The binary logistic regression indicated that CHAPS/PHONOLOGY/ABR I–V latency serves as a significant predictive indicator of auditory processing difficulties in children with SSD. Performance on CHAPS yielded an exceptionally high odds ratio, confirming the importance of difficulties in daily auditory perception as an indicator of processing disorder. PHONOLOGY scores indicated a significant negative effect, suggesting that a reduction in phonological ability is directly related to the presence of auditory processing difficulties in children with SSD. ABR wave I–V latency had yielded a low, yet significant odds ratio, affirming the neurophysiological correlation of these difficulties. Factors 5 and 13 were significantly inter-correlated. The lack of correlations among other factors highlights the relative independence of the phenomenological and neurophysiological manifestations of the disorder, reinforcing the multifactorial nature of APD, as documented by Bellis [83] and Moore et al. [84]

Allen and Allan [48] compared ABR findings in two groups of children referred for low academic performance: children diagnosed with APD and children who did not fit the APD criteria. They reported no latency differences and a lower V-I amplitude ratio in children diagnosed with APD. Their control group consisted of children with low academic achievement who did not fit the APD criteria, which may have allowed concomitant difficulties to influence findings. Our interwave I–V latency differences between the groups have been reported in other studies [53,85]. We suggest that when comparisons are made with a control group without possible coexisting challenges, the salient characteristics of the processing disorder may emerge more clearly.

In a recent meta-analysis, Maggu, Yu, and Overath [49] reported that click-evoked ABR is not affected by Auditory Processing Disorders. However, there are several studies indicating correlations of ABR findings to auditory processing performance: a major drawback in meta-analysis is the vast variability in reported studies’ experimental and control group clinical characteristics. We recommend replication and verification of findings in children with SSD to clarify and establish the patterns observed; to explain the variations identified in our study, we compare our findings with the most relevant reports. Veeranna et al. [53] indicated that the incidence of clinically abnormal interwave intervals was significantly higher in children with suspected APD (sAPD) when compared to TD children. They reported that children with APD showed significantly reduced replicability in auditory brainstem responses (ABR) and notably delayed latencies for wave I compared to TD children. In their study, the incidence of clinically abnormal absolute latencies was significantly different in children’s sAPD when compared with TD children [53]. We found significantly shorter interwave I–V latency in children with SSD, indicating either a prolongation to wave I latency, or a shorter duration of the conduction of neural signals in the brainstem. This finding differs from the results of Veerana et al. [53], and Ankmnal-Veeraanna et al. [85] who reported prolonged I–V intervals in children with APD. Their observation was interpreted as evidence of a synaptic pattern of abnormalities in children with APD or suspected APD. Our findings may either indicate a synaptic abnormality in the auditory nerve or a heightened neural excitability at the level of the brainstem. Increased neural excitability variance in children with SSD was reported for several cortical, subcortical, and cerebellar areas [86]. Effective synchronization of neural activity across various regions of the auditory system facilitates speech perception and the development of phonological skills [87]. In this study [87], ABR amplitudes did not differ significantly between the groups, consistent with our study. Similarly, there were no significant differences in the absolute latencies of waves I, III, and V, which also aligns with our findings. Ankmnal-Veeraanna et al. [53] reported that children with APD had significantly lower wave V amplitudes compared to typically developing children, whereas we found no differences in wave V amplitude between children with SSDs and TD. Both studies by the Ankmnal-Veeraanna group investigated children diagnosed with APD vs. TD children and documented that synaptic function in the lower auditory brainstem, as reflected by wave V amplitude, is significantly lower in children diagnosed with APD [53,85]. Our experimental group, children with SSD, exhibited lower performance on auditory tests, but not at a level of APD diagnosis; this functional differentiation was not reflected in the electrophysiological index of wave V amplitude. The difference in I–V interval latency may reflect a specific characteristic of this population and needs to be investigated by further research.

Auditory pathway electrophysiology in children with phonological disorders is notably different from that in typically developing children [87,88]. These studies reported that children with phonological disorders tended to show increased latencies in auditory ABR, indicating distinct operational characteristics of the brainstem auditory pathways.

Omidvar et al. [89] and Morlet et al. [90] found no significant differences in auditory brainstem response (ABR) latencies and amplitudes between children with APD or at risk of APD and TD children, consistent with our study findings. Omidvar et al. [54] noted a notable difference in click-evoked ABR wave I among children with listening difficulties. The amplitude of wave I was lower in children with listening difficulties indicating diminished neural synchrony in response to auditory stimuli at the cochlear level. Increased variability in neural responses to speech sounds may detrimentally impact children’s speech perception abilities in noisy environments. Jafari et al. [91] observed that atypical encoding of consonants and vowels at the brainstem level might lead to impaired speech perception abilities. This finding appears to be confirmed by our study, as many ABR measurements are related to phonological assessments at the syllable and phoneme levels. Furthermore, Purdy et al. [92] discovered that children with learning disabilities and APD exhibited shorter ABR latencies in wave V and III–V and lower amplitudes in wave V compared to TD peers. Conçalves et al. [93] indicated that the latencies of waves I, III, and V were longer, albeit within normal limits, in children with phonological disorders. Maggu, Yu, and Overath [49], reported no significant differences in amplitudes of ABR waves between children with and without auditory processing difficulties; however, they did report differences in the latencies of waves I, III, and V. The authors emphasized the necessity of electrophysiological assessment for accurate auditory processing diagnoses. It is posited that mild cortical dysfunctions in the left posterior superior temporal lobe may account for the phonological challenges and the lower performance on dichotic digit tasks observed in children with auditory processing and SSDs [58,87].

There is a considerable body of literature supporting strong correlations between APD and electrophysiological measures, especially with ABR interwave latencies. Further investigation of larger groups of children with clearly defined language, cognitive, and developmental characteristics and systematic AP evaluation protocols would document these correlations and identify neural pathway dysfunctions in children with APD.

### 4.3. Limitations and Future Extensions

One limitation of our study is the relatively small sample size, which may limit the generalizability of our findings. As a result, the cut-off points should not be used for diagnostic purposes as they reflect the performance of children with SSD; they should only be used in that context, and upon verification and validation with larger group sizes. The cut-off values we present may be viewed as indicative levels for concern. Future research will improve robustness by using a larger sample and by recording extensive medical history and factors such as prematurity and cognitive disorders when evaluating children with listening difficulties/suspicion of auditory processing dysfunction, as reported by Moore et al. [94]. Knowledge of medical risk factors mentioned in [94] is very useful in determining causative relationships: our study targeted a clinical/functional approach where we are pointing out areas of concern and assessment to clinicians to guide a thorough evaluation and intervention plan for children with a history of SSD.

## 5. Conclusions

This research constitutes an observational and comparative analysis of the auditory processing profiles, language capabilities, and electrophysiological indices of Greek Cypriot children aged 7 to 10 years: children with SSDs compared to their TD peers. Our study revealed notable differences across the electrophysiological parameters of ABR, and significant correlations among auditory processing indices related to phonology and auditory performance. This was the first investigation conducted among Greek Cypriot children, seeking identification of potential auditory processing difficulties in children with SSD. It serves as a foundational reference for speech and language pathologists and other members of the interdisciplinary team. To ensure accurate and comprehensive diagnosis, children with phonological disorders should undergo auditory processing evaluation assessment. Our findings indicate that when SLPs assess children with speech sound deficits, they might benefit from considering the potential effects of auditory processing deficits on language skills. Effective assessment may thereby lead to the development of intervention programs specifically tailored to the children’s difficulties.

## Figures and Tables

**Figure 1 audiolres-15-00119-f001:**
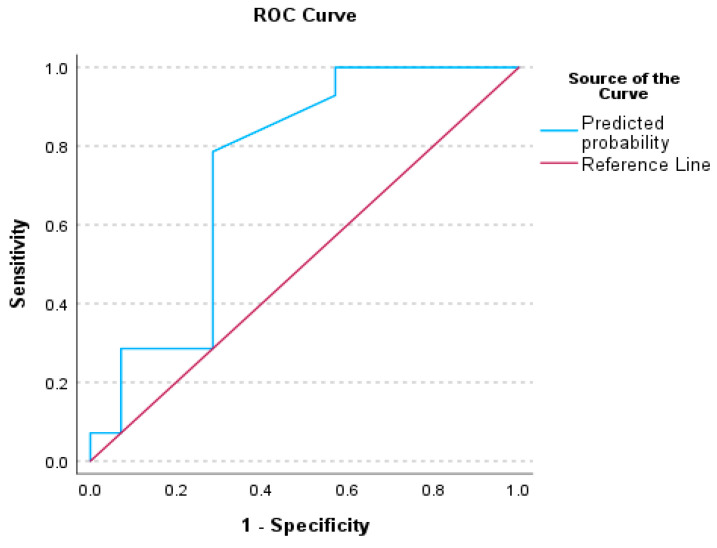
Receiver Operating Characteristics (ROC) curve for the Factor 1—CHAPS measurement—between TD children and children with SSD.

**Figure 2 audiolres-15-00119-f002:**
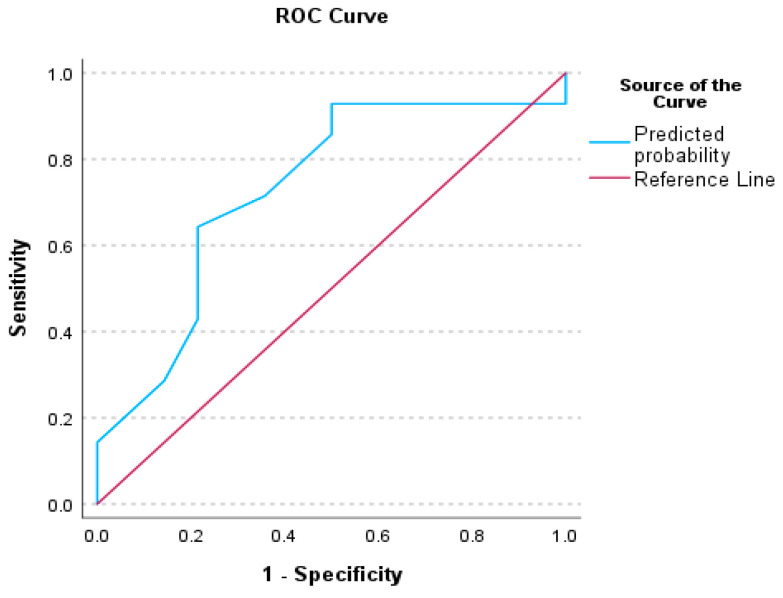
Receiver Operating Characteristics (ROC) curve for the Factor 5—PHONOLOGY between TD children and children with SSD.

**Figure 3 audiolres-15-00119-f003:**
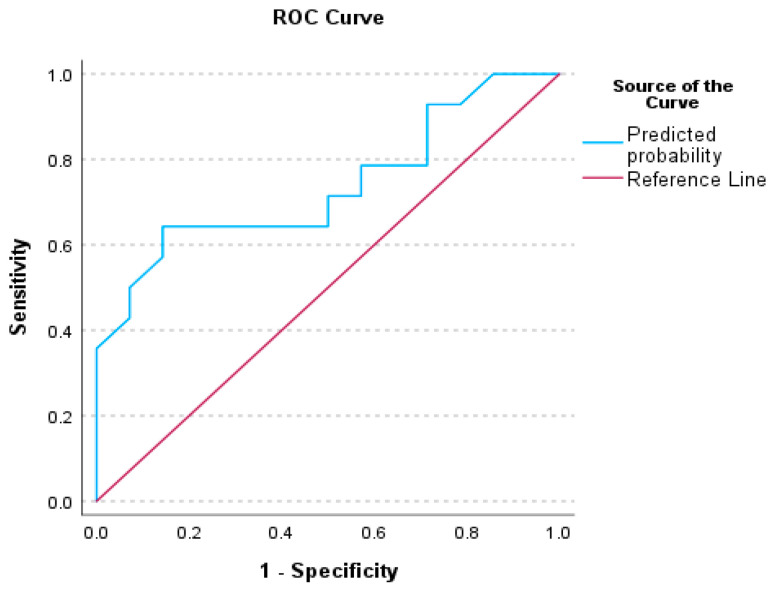
Receiver Operating Characteristics (ROC) curve for the Factor 13—ABR-wV measurement—between TD children and children with SSD.

**Figure 4 audiolres-15-00119-f004:**
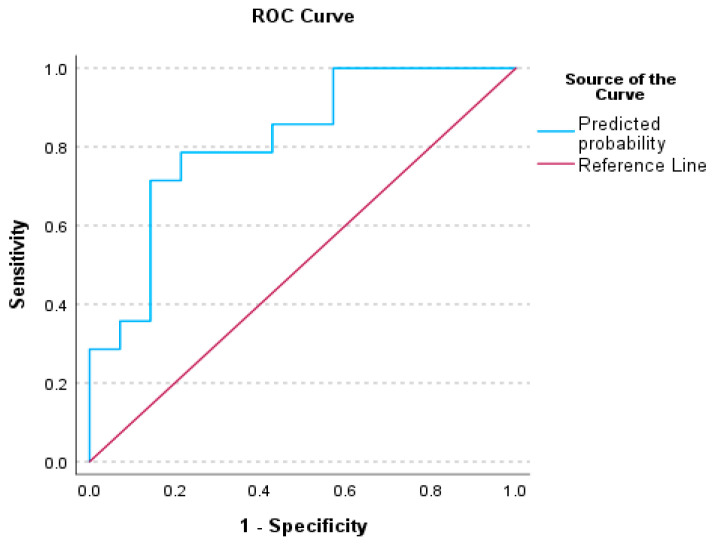
Receiver Operating Characteristics (ROC) curve for the Factor 1—CHAPS & Factor 13—ABR-wV measurement—between TD children and children with SSD.

**Table 1 audiolres-15-00119-t001:** Electrophysiological assessment parameters.

Test	Stimulus Type	Stim. Intensity	Presentation Rate	Filter
ABR	Click	60 dBnHL	37.7/s	10–1500 Hz
MLR	Click	60 dBnHL	5.7/s	10–100 Hz
LLR	1000 Hz tone	60 dBnHL	1.1/s	1–15 Hz

**Table 2 audiolres-15-00119-t002:** Chronbach’s α of significant factors.

Factors	Cronbach α
Factor 1—CHAPS	0.60
Factor 5—PHONOLOGY	0.779
Factor 13—ABR-wV	0.821

**Table 3 audiolres-15-00119-t003:** Binary Logistic Regression Models for the 3 factors.

	χ^2^	*p*	R^2^	Wald	OR	95%CI	Model Prediction
Factor 1—CHAPS	5.754	0.016	0.186	4.539	29.276	1.310–654.094	71.4%
Factor 5—PHONOLOGY	5.111	0.024	0.167	4.129	0.577	0.339–0.981	67.9%
Factor 13—ABR wV	5.209	0.022	0.170	4.094	1.0 × 10^−4^	2.0 × 10^−8^–0.753	60.7%

**Table 4 audiolres-15-00119-t004:** Spearman’s and Pearson’s Correlations between the 3 factors.

Correlations	r	*p*
Factor 1—CHAPS and Factor 13 ABR wV	−0.141 *	0.475
Factor 1—CHAPS and Factor 5—PHONOLOGY	−0.251 **	−0.197
Factor 5—PHONOLOGY and Factor 13—ABR wV	0.385 **	0.043

Abbreviations: * Pearson correlation; ** Spearman rank correlation.

**Table 5 audiolres-15-00119-t005:** ROC data analysis for variables indicating significant differences between the groups.

	AUC	AUC (95%CI)	*p*
Factor 1—CHAPS	0.740	(0.545–0.935)	0.016
Factor 5—PHONOLOGY	0.730	(0.535–0.924)	0.021
Factor 13—ABR wV	0.737	(0.547–0.928)	0.015
Factor 1—CHAPS & Factor 13—ABR-wV	0.816	(0.657–0.975)	<0.001

Abbreviations: AUC, Area Under Curve; *p* < 0.005.

## Data Availability

The data presented in this study are available on request from the corresponding author due to ethical and privacy reasons.

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
