# Peer review of "Auditory Processing and Speech Sound Disorders: Behavioral and Electrophysiological Findings"

_audiolres, 2025, doi:10.3390/audiolres15050119_

Round 1

Reviewer 1 Report

Comments and Suggestions for Authors

The topic of the article is interesting and relevant. Auditory processing disorders (APD) may lead to or be associated with difficulties in learning, speech, language, attention, social and related functions, and communication disorders. There is no single diagnostic standard for APD identification in different countries. Algorithm of examination of such patients is constantly improving. The presented research is devoted to assessment of auditory processing in 14 children with Speech Sound Disorders (SSD) compared to 14 typically developing (TD) children. The authors have done a huge job using a variety of psychophysical tests (dichotic speech, auditory discrimination, auditory frequency and duration pattern recognition, and monaural low redundancy speech tests), and electrophysiological investigations (Auditory Brainstem Response, Middle Latency Response, and Late Latency Response).

Considering the results of the research, the conclusion about the notable differences across the electrophysiological parameters, which exhibited moderate correlations with psychophysical assessments in SSD- and TD-children, is logical. The theoretical reasoning and research procedures are adequately presented. As far as I can tell from the information available the data are analyzed appropriately. Overall, the work reported in the paper represents a sufficiently significant extension of our knowledge to warrant publication.

The paper is easy to read. The title and the abstract are clear and helpful to reveal the purpose of the article. The presentation of the rationale of the work is logical and clear.

However, there are some comments:

·         P. 7, 3.3.1. Receiver Operating Characteristics (ROC) Analysis: “ROC curves were computed for the behavioral tests that showed statistical significance between the two groups as published in Drosos et al.2024b [],...” - the link’s number is missing.

·         P. 16, 3rd paragraph: “Duquette-Laplante et al. [95] reported a significantly lower performance on the Dichotic Digits task for the right ear among children suspected of having APD when compared to what”- the sentence is unclear.

·         P. 19-20. Conclusion. It would be appropriate to shorten the conclusion, reflecting only the main results

Recommendation

The paper is suitable for publication after minor revision.

Author Response

Dear reviewer 1,

We appreciate your constructive suggestions and guidance to improve the presentation of our work. We have completed the suggested changes, as detailed below:

7, 3.3.1. Receiver Operating Characteristics (ROC)Analysis:“ROC curves were computed for the behavioral tests that showed statistical significance between the two groups as published in Drosos et al.2024b [],...” - the link’s number is missing.

Response: We thank Reviewer #1 for his suggestion.

We have added the missing bibliographic reference and apologize for the initial oversight.

16, 3rd paragraph: “Duquette-Laplante et al. [95] reported a significantly lower performance on the Dichotic Digits task for the right ear among children suspected of having APD when compared to what”- the sentence is unclear.

Response: We are grateful for your attention to this matter; we have revised the sentence to clarify its meaning.

19-20. Conclusion. It would be appropriate to shorten the conclusion, reflecting only the main results

Response: Thank you for the recommendation.

We have revised the conclusions, and we have included the main results.

Reviewer 2 Report

Comments and Suggestions for Authors

The authors studied auditory processing profiles in children with speech sound disorders. It was an interesting study and the findings may help the health professionals in managing the patients with the disorder. The methodology was adequately written and the results were well presented. The introduction, discussion and conclusion were satisfactorily done.

Minor comments:

1. To state the reason of using p value as 0.005. Usually the p value of 0.05 is good enough.

2. To state the actual p value rather than just stated the p value of <0.005.

3. The conclusion does not need citation.

Author Response

Dear Reviewer 2,

Thank you for your valuable comments and suggested revisions. We have completed the suggested changes, as detailed below:

To state the reason of using p value as 0.005. Usually, the p value of 0.05 is good enough.

Response: Thank you for pointing it out, the p-value typing error has been corrected.

To state the actual p value rather than just stated the p value of <0.005.

Response: Thank you for your comment, the actual p-value was stated where needed.

The conclusion does not need citation.

Response: The conclusions have been condensed to include only the main findings, and the bibliographic references have been removed.

Reviewer 3 Report

Comments and Suggestions for Authors

Variables should be named properly (e.g., ABR_RIGHT_IPSILATΕRAL_V_MS etc.).

I believe, sample size is small for ROC analysis. How the sample size was calculated?

Behavioral thresholds are not significantly different between groups as reported in table 2.

In method for recording AEPs, more description is required as it is difficult to understand the meaning of contralateral amplitude.

In results, is it necessary to adjust p values for variables of AEPs? e.g., latency of wave V, since the comparison is carried out only once (i.e., between groups) for the latency of wave V.

Author Response

Dear reviewer 3,

Thank you for the constructive comments. We have modified the text accordingly where suggested.

Variables should be named properly (e.g., ABR_RIGHT_IPSILATΕRAL_V_MS etc.).

Response: Thank you for your comment.

All Variables are now named properly.

I believe, sample size is small for ROC analysis. How the sample size was calculated?

Response: Thank you for your valuable comment.

The sample size in our study was calculated utilizing the EPI INFO software developed by the WHO for the field of psychiatry, with a confidence interval of 95% and an expected SSD prevalence of 1,7% among the school-aged population as indicated in the Cyprus Ministry of Education, Youth and Sports (2016), and an accuracy of 3%. The sample size calculation yields 14 children per group, totaling 28 children. Reference: https://www.cdc.gov/epiinfo/pc.html

Behavioral thresholds are not significantly different between groups as reported in table 2.

Response: Thank you for your comment.

Audiometry thresholds did not differ in our study, as normal hearing thresholds were an inclusion criterion. Linguistic, cognitive and academic comorbidities were exclusion criteria for participants. The behavioral tests on which our sample had differences are reported in https://www.mdpi.com/2039-4349/14/4/53  in tables 1 and 2.  The TD and SSD group linguistic and behavioral auditory processing characteristics and statistically significant differences are detailed in these tables.

Table 1. Sample characteristics and language performance.

TD Group
(N = 16)

Experimental Group (N = 24)

Mdn (IQR)

Mdn (IQR)

p

n2

Children’s Age in Years

8.40 (7.85–8.70)

8.10 (7.90–8.40)

0.486

0.007

Raven Total Score

33.00 (31.50–33.50)

32.00 (31.00–33.00)

0.153

0.004

Action Picture Test Total Score

71.00 (62.50–76.00)

68.00 (60.00–71.00)

0.170

0.102

Action Picture Test: information

37.00 (32.00–38.50)

33.00 (31.00–36.00)

0.930

0.007

Action Picture Test: grammar

34.00 (30.50–36.50)

30.00 (28.00–32.00)

<0.005 *

0.150

Metaphone Total Score

180.00 (166.50–187.50)

169.00 (163.00–175.00)

0.220

0.099

Words from Same Syllable Score

13.00 (12.00–13.00)

13.00 (12.00–13.00)

0.936

0.002

Syllabication Score

8.00 (8.00–8.00)

8.00 (8.00–8.00)

0.830

0.006

First Syllable Identification Score

12.00 (12.00–12.00)

12.00 (12.00–12.00)

0.936

0.000

Final Syllable Localization Score

24.00 (23.00–26.00)

24.00 (22.00–26.00)

0.630

0.006

Word Finding Syllable Criterion Score

8.00 (7.00–8.00)

8.00 (8.00–8.00)

0.708

0.006

First Phoneme Identification Score

27.00 (24.50–28.50)

23.00 (22.00–26.00)

<0.005 *

0.216

Word Finding Phoneme Criterion Score

15.00 (13.50–17.00)

14.00 (12.00–15.00)

0.099

0.071

Word Repetition Score

37.00 (34.50–39.50)

34.00 (32.00–37.00)

<0.005 *

0.075

Abbreviations: TD group, typical developing children; Mdn, Medians; IQR, interquartile range; * p level at p < 0.005; n2, partial eta squared.

Table 2. Auditory processing performance.

TD Group
(N = 16)

Experimental Group (N = 24)

Mdn (IQR)

Mdn (IQR)

p

n2

Gap Detection 500 Hz

10.00 (10.00–10.00)

10.00 (10.00–10.00)

0.147

0.053

Gap Detection 1000 Hz

10.00 (10.00–10.00)

10.00 (10.00–10.00)

=0.005

0.095

Gap Detection 2000 Hz

10.00 (10.00–10.00)

10.00 (10.00–10.00)

0.809

0.001

Gap Detection 4000 Hz

10.00 (10.00–10.00)

10.00 (9.25–10.00)

0.127

0.059

Gap Detection in Noise 1

7.00

(7.00–7.00)

7.00 (7.00–7.00)

0.147

0.053

Gap Detection in Noise 5

6.00

(5.00–7.00)

5.00 (2.25–5.75)

<0.005

0.195

Speech in Bubble Right Ear

64.50 (58.25–66.00)

54.00 (39.75–62.00)

<0.005

0.143

SNR 7

70.00 (60.00–78.75)

60.00 (51.25–73.75)

0.165

0.049

SNR 5

67.50 (55.00–70.00)

60.00 (45.00–73.75)

0.284

0.029

SNR 3

65.00 (55.00–70.00)

55.00 (40.00–63.75)

=0.005

0.093

SNR 1

62.50 (55.00–70.00)

60.00 (50.00–65.00)

0.151

0.052

SNR Minus 1

50.00 (36.25–55.00)

40.00 (20.00–45.00)

<0.005

0.156

SNR 50%

2.40

(1.85–2.95)

3.00 (2.25–4.40)

=0.005

0.092

Dichotic Words—Right Ear 1st

21.00 (21.00–22.00)

19.50 (18.25–21.00)

<0.001

0.336

Dichotic Words Synchronized

21.00 (20.00–22.00)

18.00 (16.25–21.00)

=0.001

0.264

Duration Pattern Sequence

30.00 (29.50–30.00)

28.00 (26.25–29.00)

<0.001

0.323

Pitch Pattern 1000 to 2000 Hz

8.00 (7.25–8.00)

7.50 (7.00–8.00)

0.067

0.086

Pitch Pattern 4000 to 6000 Hz

8.00 (8.00–9.00)

7.00 (7.00–8.00)

=0.001

0.269

Forward Digit Span

8.00 (7.00–9.00)

7.00 (7.00–8.00)

0.200

0.042

Backward Digit Span

6.00 (5.00–7.00)

5.00 (4.25–6.00)

<0.005

0.108

Abbreviations: TD group, typical developing children; Mdn, Medians; IQR, interquartile range; SNR, signal-to-noise ratio; p level at p < 0.005; n2, partial eta squared.

In method for recording AEPs, more description is required as it is difficult to understand the meaning of contralateral amplitude.

Response: We have added a more detailed description of AEPs in Methods. Research in auditory evoked potentials (AEPs) indicates that auditory cortex maturation manifests as modified hemispheric asymmetry. The two hemispheres may support distinct auditory processing specializations: these asymmetries were assessed with ipsilateral and contralateral  AEP recordings. Ipsilateral AEP recording entails recording of electrical activity in response to auditory stimuli delivered to the same ear; latencies and amplitudes were calculated for both the ipsilateral and contralateral recordings of ABR, MLR, LLR. This methodology  has been applied to evaluate the overall functionality of the auditory system, with a particular focus on the integrity of the neural pathways involved [22,29].

In results, is it necessary to adjust p values for variables of AEPs? e.g., latency of wave V, since the comparison is carried out only once (i.e., between groups) for the latency of wave V.

Response: Thank you for your comment. The measurements of AEPs include absolute and inter-wave latencies; a delay in wave I latency in the cochlea may consequently affect wave V latency.  Indeed, for the MLR and LLR we did not need to adjust the p values, but we opted to adjust the p-values to comprehensively assess the factors associated with ABR, MLR, and LLR, considering the more conservative power resulting from the adjustment.

Reviewer 4 Report

Comments and Suggestions for Authors

Major revisions,

The symptoms of SSD experienced by the subjects in this study are unclear, and the analysis method needs to be clarified. It can also be said that the fundamental concept of APD need to be reconsidered including recent study findings.

1)     Could you show more detail information of participants? The participants has been diagnosed with SSD, but authors did not indicate their symptoms, the severity, and the diagnostic criteria. The tests shown in Tables 1 and 2 do not show any differences between SSD and TD children. Which symptoms led to the diagnosis of SSD? Please also indicate the exclusion criteria for participants.

2)     Please show the details of the tests shown on page 4, such as Gap Detection, Gaps-in-Noise, Dichotic Hearing (words). As the methods used differ from country to country, the careful explanation is required.

3)     The Electrophysiology Assessment Findings are very difficult to understand, but does this mean that there were no differences in the right ear and only in the left ear? It is difficult to interpret why there is such a difference in brainstem response between the left and right ear. Recent studies showed the cause of APD are cognitive problems. From this perspective, the findings of your study are incomprehensible, and it may indicate that the participants of your study may have some other problem. Authors should read the recent studies and organize author’s hypothesis. For example, Moore et al (2024) https://www.medrxiv.org/content/10.1101/2024.10.02.24314796v1.full.pdf

4)     Despite the fact that there are no difference between the two participant groups (Table1 and 2), authors conducted ROC analysis. The ROC analysis is meaningless when there are no differences between the two groups and the number of participants is very small. The specificity of the results is also low. These results show that the ROC analysis itself was not appropriate for your study. The analysis method needs to be examined.

5)     Other results, such as correlations and multiple comparisons, are also shown in your manuscript, but the results (Table 5-11) are complicated and need to be organized and presented, including the fundamental theory.

6)     Authors need to reconsider the analysis methods and reorganize your thoughts accordingly in discussion.

Author Response

Dear reviewer 4,

Thank you for your valuable comments and suggested revisions. We have completed the suggested changes, as detailed below:

 The symptoms of SSD experienced by the subjects in this study are unclear, and the analysis method needs to be clarified. It can also be said that the fundamental concept of APD need to be reconsidered including recent study findings.

Response: Thank you for your valuable comment.  Participants selected to enroll in the SSD group had a history of SSD diagnosis based on the criteria of ICD-10, code F80.1, followed by Speech Therapy intervention. Intervention had been concluded by the time of participation in our study.   

Could you show more detail information of participants? The participants has been diagnosed with SSD, but authors did not indicate their symptoms, the severity, and the diagnostic criteria. The tests shown in Tables 1 and 2 do not show any differences between SSD and TD children. Which symptoms led to the diagnosis of SSD? Please also indicate the exclusion criteria for participants.

Response: Thank you for your comment. As indicated in https://pmc.ncbi.nlm.nih.gov/articles/PMC11270421/ children with SSD diagnosis had significant differences in Language behavioral performances as they differed in grammar, first phoneme identification, and word repetition. Children with SSDs showed significantly lower performance in Auditory processing behavioral tasks specifically in Gap Detection in quiet and noise, Speech in Babble, Dichotic Words, Duration and Frequency Patterns, and backward digit span repetition. The Frequency Pattern Test scores were significantly correlated with Rhyme Identification. We have made corrections and removed Tables 1 and 2 from our results. Additionally, we have noted the reference article that describes in detail the behavioral assessments which were conducted.

Please show the details of the tests shown on page 4, such as Gap Detection, Gaps-in-Noise, Dichotic Hearing (words). As the methods used differ from country to country, the careful explanation is required.

Response: Thank you for your recommendation. We have included the protocol content with additional details regarding the audiological tests:

The behavioral test protocol was also presented in [59]. For the sake of completeness, we present the protocol here. A pure-tone audiogram was administered by an audiologist at the Speech, Language, and Hearing Clinic of the European University Cyprus [64]. Consequently, the children completed the auditory processing evaluation comprising of the following tests: Gap Detection [65], Gaps-in-Noise [66], Dichotic Hearing (words) [67], the Greek Speech in Babble [68], Duration and Frequency Pattern Sequence [68], and Forward/Backward digit span [69]. Gap Detection and Gaps in Noise incorporate non- linguistic stimuli, following the implementation protocol established by Musiek (1994). The Dichotic Listening and Speech in Babble test administered has been validated for the Greek language (Iliadou and Bamiou, 2012; Iliadou et al., 2016). Gap detection entails detecting a short silence or pause between two sounds within a continuous auditory stream (Musiek, 1994). Gaps in Noise refers to the ability to detect a brief silent interval or gap within a background of continuous noise. Gap detection in noise is often used in auditory processing research to evaluate how well individuals can perceive temporal cues when they are masked by ongoing, potentially distracting background sounds (Musiek,1994). Dichotic listening (words) involves the simultaneous presentation of different words to each ear. The goal is to assess an individual's ability to process and integrate sounds coming from both ears, as well as to evaluate their auditory attention and lateralization abilities. Dichotic tasks assess how well each hemisphere of the brain processes auditory information and show the hemispheric specialization and cross-communication (Iliadou and Bamiou,2012; Iliadou et al., 2016).

The Electrophysiology Assessment Findings are very difficult to understand, but does this mean that there were no differences in the right ear and only in the left ear? It is difficult to interpret why there is such a difference in brainstem response between the left and right ear. Recent studies showed the cause of APD are cognitive problems. From this perspective, the findings of your study are incomprehensible, and it may indicate that the participants of your study may have some other problem. Authors should read the recent studies and organize author’s hypothesis. For example, Moore et al (2024) https://www.medrxiv.org/content/10.1101/2024.10.02.24314796v1.full.pdf

Response: Thank you for your comment. It is important to note that all audiological and electrophysiological tests were conducted to document the auditory processing characteristics of the sample of children with SSD. The aim of this study was to identify potential AP differences between children with SSD and TD peers: there was no intention to diagnose APD. Additionally, our objective was to record any correlations between linguistic indices, behavioral AP test performance, and electrophysiological measurements. As dictated by our exclusion criteria, our sample had normal cognitive function as indicated by Raven test.  We excluded children with ADHD or learning disorders, including dyslexia.

We have incorporated into the manuscript extensive literature documenting variations in AP electrophysiological assessments between clinical groups, and between the two ears (Ankmnal-Veeranna et al. [107]; Rocha-Muniz, Befi-Lopes, Schochat [29]), regarding electrophysiological measurements.  There are also references where no differences are observed in electrophysiological AP studies (Maggu, Yu, and Overath [49]). In our study, variations were recorded in three measurements, as detailed in Table 1. This marks the first instance of such an observation in a population with SSD within a Greek Cypriot sample.

Moore et al.'s article on “listening difficulties” relationship to health risks provides very important information to the assessment process.  However, for the purposes of this study we recorded data related to current function of children with a history of a specific clinical finding (SSD) that had already been treated.  We have included the reference to the literature and the necessity of assessing these factors in children with listening difficulties to the limitations of our study. We would also like to note that our article was submitted for evaluation to the journal prior to the publication of this specific article.

Despite the fact that there are no differences between the two participant groups (Table1 and 2), authors conducted ROC analysis. The ROC analysis is meaningless when there are no differences between the two groups and the number of participants is very small. The specificity of the results is also low. These results show that the ROC analysis itself was not appropriate for your study. The analysis method needs to be examined.

Response: Our article at https://www.mdpi.com/2039-4349/14/4/53, reports significant differences between these two groups.  This report was based on larger groups that had completed the behavioral AP protocol.  For the first draft of this paper, we have included 14 children in each group, to keep the group numbers even, as only 14 children in the SSD group had completed the electrophysiological assessment. The differences observed when the larger groups were compared guided our decision to conduct ROC analyses. We have decided to include these findings in this revision, and to keep the ROC analysis.  We believe that ROC curve analysis offers significant clinical insights to the multidisciplinary team. Despite the relatively limited sample size, we opted to document all potential clinical implications. The objective of this study was to explore whether initial evidence suggests that auditory processing tests could yield valuable information for referrals to an audiologist for a thorough evaluation and for assessing the potential comorbidity with Auditory Processing Disorder.

We have modified the method and results section to address this comment, and we thank the reviewer for pointing out this serious flaw in our first submission.  We believe this comment has corrected a serious limitation and made our paper more robust.

Other results, such as correlations and multiple comparisons, are also shown in your manuscript, but the results (Table 5-11) are complicated and need to be organized and presented, including the fundamental theory.

Response: Thank you for your comment. We have removed previous Tables 5-8 to enhance the flow of the manuscript, retaining only the explanatory texts regarding the correlations and Tables 9-11 (and with the new order 4-6) which exclusively present the electrophysiological and audiological associations. Additionally, the fundamental theory has been incorporated into the statistical analysis within the Methods section.

 Authors need to reconsider the analysis methods and reorganize your thoughts accordingly in discussion.

Response: Thank you for your comment; several changes have been made to the text to clarify methods, statistical analysis, results and discussion of this work.

Round 2

Reviewer 3 Report

Comments and Suggestions for Authors

This study was carried out to compare the auditory processing profiles of children with SSD and TD children. Objectives of the study was (1). to assess sensitivity, validity, and correlation between indices and (2). identify areas of intervention. Findings of sensitivity and correlation analysis are reported. Second objective is not addressed in the manuscript.

Method

sample - paragraph 2, line 6 - "while 28 children in each group (TD, SSD) completed the electrophysiological measurements" - here 'each group' should be corrected since the number of participants are 16 and 24 in TD and Exp group.

Did you use stimulus rate of 37.7/s for recording ABR, MLR, and LLR? as described in section '2.4 Electrophysiological Assessment'.

Result

"Children with SSD had ... significantly reduced performance, in ... the Speech in Babble test [67] at S/N form -1 to +7 - here p values are significant only at -1 and 3 dB SNR.

Table 1. ABR showing latency or amplitude?

what is the purpose of cutoff scores? based on cutoff score diagnosis is SSD or APD?

Author Response

Dear Reviewer 3, we would like to express our gratitude for your comments. Below, we have included your feedback along with our response, detailing the changes made to enhance the manuscript following your suggestions.

This study was carried out to compare the auditory processing profiles of children with SSD and TD children. Objectives of the study was (1). to assess sensitivity, validity, and correlation between indices and (2). identify areas of intervention. Findings of sensitivity and correlation analysis are reported. Second objective is not addressed in the manuscript.

Method

sample - paragraph 2, line 6 - "while 28 children in each group (TD, SSD) completed the electrophysiological measurements" - here 'each group' should be corrected since the number of participants are 16 and 24 in TD and Exp group.

Thank you for the comment, the number of participants in each group was corrected.

Did you use stimulus rate of 37.7/s for recording ABR, MLR, and LLR? as described in section '2.4 Electrophysiological Assessment'.

Thank you for the comment. A detailed table with the AEP recording parameters has now been included in the Methods section (Table 1, in section 2.4). 

Result

"Children with SSD had ... significantly reduced performance, in ... the Speech in Babble test [67] at S/N form -1 to +7 - here p values are significant only at -1 and 3 dB SNR.

Thank you for the comment. The form was corrected in Results in 3.1 section, first paragraph.

Table 1. ABR showing latency or amplitude?

In the ABR presented in Table 2, we observe the latency of the I-V wave, with the MS notation indicated alongside the time, as also detailed in Appendix Table 1A. The remaining points are marked with the AMPLITUDE designation.

What is the purpose of cutoff scores? based on cutoff score diagnosis is SSD or APD?

The cut off scores here are referring to Auditory Processing measures in children with SSD: they were used to verify the reliability of AP tests in identifying APD characteristics among children with a history of SSD.  Therefore, as the AP assessment protocol was used to create the curves, and the SSD was our clinical group, the significant findings in our ROC analysis indicated the significant presence of APD among children with SSD.

Reviewer 4 Report

Comments and Suggestions for Authors

Thank you for considering the revised manuscript. I reviewed it, but it has serious problems that cannot be solved by a short-term revise. Therefore, I am unable to recommend this paper for publication in its current form.

1)     Selection of participants

 If the authors examine the differences between participants with SSD and TD, authors have to show the difference of participant groups. However, authors showed only the information that participants with SSD had previously been diagnosed with SSD, and no other information of their symptoms and severity of SSD. The exclusion criteria is also unclear. It is difficult to interpret the results of SSD children with minimal information.

2)     The interpretation of AP tests

 The interpretation of the results in the AP test is not consistent with other recent findings. It has been suggested that various cognitive abilities are involved in the AP test, and it is not possible to conclude that SSD children have a decline in AP ability based on the results of the AP test. As there are no details of the participant’s information, it is also unclear what this study examined.

3)     The electrophysiological tests

There is no explanation of basic question why there was a difference in only one ear. The authors did not interpret the detailed data.

4)     ROC analysis

I pointed out that ROC analysis is meaningless for data with a small sample size and no significant differences between participant groups, but the author’s response was that it was based on the differences in previous studies. The result of previous study is not a reason for the analysis of the current study.

From the above, authors have to consider the participant’s information, analysis and the interpretation of results. I think that authors need to rewrite the manuscript and resubmit it.

Author Response

Dear Reviewer 4, we would like to express our gratitude for your comments. Below, we have included your feedback along with our response, detailing the changes made to enhance the manuscript following your suggestions.

Thank you for considering the revised manuscript. I reviewed it, but it has serious problems that cannot be solved by a short-term revise. Therefore, I am unable to recommend this paper for publication in its current form.

We would like to emphasize that we have addressed all the issues you raised with respect and careful consideration. We kindly ask you to review the revised manuscript .

1)     Selection of participants

 If the authors examine the differences between participants with SSD and TD, authors have to show the difference of participant groups. However, authors showed only the information that participants with SSD had previously been diagnosed with SSD, and no other information of their symptoms and severity of SSD. The exclusion criteria is also unclear. It is difficult to interpret the results of SSD children with minimal information.

The children in our experimental group had a history of SSD diagnosis and related treatment that had been concluded at the time of participation in our study.  The fact that all children with SSD had been treated indicated a level of the disorder that necessitated intervention, thus a significant level warranting the SSD diagnosis. In our study, the children exhibited mixed phonological and articulation disorders. They did not present cognitive, learning, or other difficulties such as attention deficit, which are exclusion criteria for our research. Furthermore, the screening tools CHAPS and APDQ administered did not reveal any challenges related to attention. The behavioral assessments included a test for the recall of numerical digits. In this particular test, our sample showed differences in reverse recall, and this finding has been documented and discussed in the text, as well as in the introduction, results, and correlations between behavioral and electrophysiological measurements. 

The exclusion criteria warranted participation of children with normal hearing and cognitive functions, to fit the requirements for AP assessment. 

We followed the inclusion and exclusion criteria reported in the literature, to eliminate potential cognitive difficulties such as memory and attention disorders, as well as learning difficulties. For this reason, we obtained a comprehensive history, assessed hearing, administered the RAVENS test, conducted a double-check using screening tools that include subdomains related to attention, and administered digit memory tests (both forward and backward recall).  This approach ensured that cognitive difficulties were not present. The article proposed to us regarding medical factors associated with auditory processing disorders and listening disorders emphasizes the examination of memory and attention, which we have also included; however, we did not use the same evaluative tool as it is not standardized in the Greek language. 

2)     The interpretation of AP tests

 The interpretation of the results in the AP test is not consistent with other recent findings. It has been suggested that various cognitive abilities are involved in the AP test, and it is not possible to conclude that SSD children have a decline in AP ability based on the results of the AP test. As there are no details of the participant’s information, it is also unclear what this study examined.

To account for cognitive function and ascertain that cognitive function would not be a confounding factor, we included the RAVEN matrices in our protocol: the RAVEN matrices provide a valid nonverbal means of assessing cognitive function.  Typically Developing children performed similarly to children with a history of SSD, therefore we can expect that overall cognitive function was of similar level in the two groups.   

It is also important to note that similar methodologies were employed in other articles on the subject (Vilela et al. 2016, Barozzo et al. 2016, Jain et al. 2019; Jain et al. 2024) to assess behavioral factors. An additional aspect they included was the evaluation of memory and attention, particularly due to the challenges faced by the sample diagnosed with specific learning difficulties in the case of Jain et al. (2024). In contrast, the other articles focus on specific factors and tests related to auditory processing and phonology. In our study, we have implemented a more comprehensive protocol and excluded children with learning difficulties and attention disorders from our criteria.

The article Medical Risk Factors Associated with Listening Difficulties in Children. https://www.medrxiv.org/content/10.1101/2024.10.02.24314796v1.full.pdf was published after this work was completed. The reported research used the NIH Toolbox Cognition Battery, which includes up to eight standardized cognitive tests. In that study, participants completed four tests: Picture Vocabulary test, Flanker Inhibitory Control and 189 Attention test, Dimensional Change Card Sort test, and Picture Sequence Memory test. As this tool has not been validated in the Greek language, we have addressed the factors of memory and attention, which should be taken into consideration.

3)     The electrophysiological tests

There is no explanation of basic question why there was a difference in only one ear. The authors did not interpret the detailed data.

Our work focused on investigating the possibility of APD characteristics among children with history of SSD.  As the behavioral assessment indicated significant comorbidity, we are comparing our findings with literature referring to children diagnosed with or suspected of APD: we are supporting a cautionary approach when relating findings.

ABR findings

The meta-analysis [49] indicated absence of ABR characteristics’ differentiation of APD.  Conversely, Veraana et al., [53] indicated that there may be some variations in the electrophysiological measurements, (see discussion in the present paper). In our study, we identified a significant difference in the Left Ear I-V interwave latency as presented in Table 1.  This may indicate that children with SSD may exhibit prolonged processing time along the VIIIth cranial nerve, namely between the distal part (main contributor to wave I) and the Lateral Lemniscus/Inferior Colliculus (main contributors to wave V).  Veerana et. all [108] indicated that “incidence of clinically abnormal interwave intervals was significantly higher in children with suspected APD (sAPD) when compared TD children”, as they found significant interwave I-III &III-IV differences between TD and sAPD children.   Our finding falls along these lines of discovery.  The clinical population here are the children with SSD, therefore the difference in I-V interval latency may reflect a specific characteristic for this population and needs to be investigated by further research. 

MLR and LLR findings

Macaskill et al. [119] reported significant differences in waves N1 and N2 between children with APD and normal hearing peers.  They recommended than N1 and/or N2 can be considered as biomarkers for APD.  Mattsson et al. [26] indicated that amplitude of LLR wave N1 differed between TD children and children with APD diagnosis, whereas our work indicated a higher P1 amplitude in our experimental group.  As P1 reflects early cortical responses and N1 engages broader cortical networks, we might consider the possibility that the auditory processing characteristics observed in children with a history of SSD correspond to earlier cortical function differences.     

4)     ROC analysis

I pointed out that ROC analysis is meaningless for data with a small sample size and no significant differences between participant groups, but the author’s response was that it was based on the differences in previous studies. The result of previous study is not a reason for the analysis of the current study.

The minimum group size requirement for statistical power was fulfilled even with the reported relatively low group sizes.  The publication reporting significant differences in behavioral measures was based on data from the same study.  In that publication (Preliminary Validation of the Children’s Auditory Performance Scale (CHAPS) and the Auditory Processing Domain Questionnaire (APDQ) in Greek Cypriot Children

https://www.mdpi.com/2039-4349/14/4/53), we analyzed the validity and reliability of the screening questionnaires. However, the group size for the electrophysiology results comparisons was dictated by the fact that fewer children completed the electrophysiological assessment.  

The ROC curve analysis was used to verify the reliability of AP tests in identifying APD characteristics among children with a history of SSD.  Therefore, as the AP assessment protocol was used to create the curves, and the SSD was our clinical group, the significant findings in our ROC analysis indicated the significant presence of APD among children with SSD.

In conclusion:

 We respectfully submit that this manuscript contributes significantly to the narrative of APD and language disorder comorbidities; our findings concur with literature reports, and strongly support the sensitization of the multidisciplinary team to potential Auditory Processing difficulties among children with SSD: we believe that assessing AP function in children with a history of SSD and persistent academic difficulties can reinforce and focus the intervention process to optimize valid personalized treatment approaches.